# Fourier Features Let Agents Learn High Precision Policies with Imitation Learning

## Abstract

Various 3D modalities have been proposed for high-precision imitation learning tasks to compensate for the short-comings of RGB-only policies. Modalities that explicitly represent positions in Cartesian space, such as most point cloud encoder architectures, have an inherent advantage over purely image-based ones, since they allow policies to reason about geometry. Despite the effectiveness of such architectures, a number of hybrid 2D/3D architectures have been proposed in the literature, indicating that this performance can often be task-dependent. We hypothesize that this discrepancy may be due to the spectral bias of neural networks towards learning low frequency functions, which especially affects architectures conditioned on slow-moving Cartesian features. We thus propose to use a parametric projection to map point clouds from Cartesian space into high-dimensional Fourier space when using a point cloud encoder. We experimentally validate the use of these Fourier features on challenging manipulation tasks from the RoboCasa and ManiSkill3 benchmarks, and on a real robot setup. Despite their simplicity, we find that Fourier features provide robust and significant benefits across diverse encoder architectures and tasks. These results indicate that Fourier features let policies leverage geometric details more effectively than Cartesian features, showing their potential as a general-purpose tool for point cloud-based imitation learning. The overview and demos are available on our project page: https://fourier-il.github.io/fourier-il.

## 1 Introduction

Diffusion-based imitation learning (IL) has emerged as a powerful framework for robotic visuomotor control (Chi et al., 2023; Reuss et al., 2023; Wu et al., 2025; Intelligence et al., 2025). By treating action generation as a denoising process (Ho et al., 2020), diffusion policies naturally capture multimodal action distributions, enabling robots to represent the diverse strategies often present in human demonstrations. This capability has made diffusion policies the state-of-the-art on long-horizon and multi-task manipulation benchmarks.

Diffusion models excel at capturing the multi-modality of expert demonstrations, but require the input representations to preserve the fine-grained information that distinguishes successful strategies from failed ones. In high-precision manipulation tasks, 3D information about the scene can help the agent reason about geometry and occlusions and execute complex motions accurately. Policies that cannot perceive fine geometric information encoded in observations are unable to imitate expert demonstrations that depend on these details.

RGB images remain the most common observation space due to their semantic richness and the widespread availability of pretrained vision encoders (Ke et al., 2025; Wilcox et al., 2025; Donat et al., 2025). However, they lack explicit 3D geometry and require the policy backbone to implicitly infer a 2D-to-3D mapping, while also being sensitive to viewpoint and lighting variations. In contrast, 3D modalities such as depth maps, point clouds, or point maps that explicitly encode shape, distance, and spatial relationships, allow policies to learn behaviors in a common 3D space, enabling multi-view consistency. Yet despite the success of these 3D representations (Ze et al., 2024; Zhu et al., 2024; Ze et al., 2025), a number of hybrid 2D/3D architectures have recently been suggested (Ke et al., 2025; Wilcox et al., 2025; Goyal et al., 2023), indicating that the performance of 3D representations may depend on specific tasks and dataset.

Figure 1: **Method Overview.** Adding a Fourier feature mapping from Cartesian coordinates into a higher-dimensional feature space improves performance for any point cloud encoder used for diffusion imitation learning. For high-precision policies, the network must learn to condition on fine details in the scene geometry to e.g. device whether to insert the leg into the slot or reposition it, yet neural networks learn the high frequency components of the target function only slowly, if at all. While neighbouring points in the scene have very similar Cartesian features, the high-dimensional Fourier features allow them to easily be distinguished.

While neural networks are universal function approximators (Hornik et al., 1989), they have a *spectral bias* toward learning low-frequency components first, while high-frequency components converge slowly or may not be learned at all (Rahaman et al., 2019; Tancik et al., 2020). In the context of precise robotic manipulation tasks, such as inserting a peg into a socket, these high frequency components can make the difference between a successful trajectory and one where the peg and the socket are slightly mis-aligned. Both observations are very similar in terms of absolute Euclidean coordinates and distance, making it difficult to robustly learn to differentiate between them. In fields such as novel view synthesis, this shortcoming is remedied using a Fourier feature mapping (Mildenhall et al., 2021; Tancik et al., 2020). More recently Adapt3R (Wilcox et al., 2025) has shown that incorporating Fourier features benefits their architecture, yet a systemic study of Fourier features for other 3D representations in imitation learning is missing in the literature.

Inspired by these insights, we propose to encode the 3D representations for pointcloud-based approaches (Qi et al., 2017a; Gyenes et al., 2024) in Fourier space. By amplifying high-frequency components of these representations, we counteract spectral bias and make subtle temporal and geometric differences accessible to diffusion backbones. This simple modification allows different models acting on 3D representations to more easily understand small details in geometric observations, thus improving their performance for high-precision control tasks. Experimentally, we show that using Fourier-encoded input representations leads to consistent improvements across different point cloud architectures and benchmarks. In RoboCasa (Nasiriany et al., 2024) and ManiSkill3 (Tao et al., 2025), we achieve an average success rate improvement of up to $18\%$ and $7\%$, respectively, and an increase in average normalized score from $14.8\%$ to $40.2\%$ on 4 challenging real world tasks. Qualitatively, policies trained with Fourier mappings exhibit smoother and more precise motions, particularly on robotic control tasks where fine-grained manipulation matters.

Our contributions are the following: **1)** we introduce a framework for incorporating Fourier feature mappings into various point cloud encoders; **2)** through experiments on a real robot and on the RoboCasa and ManiSkill task suites, we demonstrate consistent and robust improvements over baselines without Fourier feature mappings; and **3)** through extensive ablations, we show that Fourier features do not require additional regularization and are robust to choice of hyperparameters.

## 2  RELATED WORK

**Imitation Learning in Robotics.** Recent progress in IL has been driven by incorporating diffusion (Chi et al., 2023; Reuss et al., 2023) or flow matching (Lipman et al., 2023), which enable policies to learn multi-modal action distributions, and by training policies on large-scale datasets (Black et al., 2024; Intelligence et al., 2025; Brohan et al., 2022; Zitkovich et al., 2023; Zhu et al., 2025) to significantly increase generalization and performance. However, these approaches are primarily conditioned on RGB images. This choice allow leveraging powerful pretrained visual encoders and provides strong semantic features, but RGB inputs lack explicit 3D geometry and are sensitive to viewpoint and lighting variations (Zhu et al., 2024; Ze et al., 2024; Wilcox et al., 2025). To address these shortcomings, several works incorporate 3D information by either entirely using 3D inputs or by combining them with RGB (Ze et al., 2024; Gervet et al., 2023; Wilcox et al., 2025; Goyal

et al., 2023). In our work, we focus specifically on using 3D inputs for high-precision manipulation tasks and show that their performance is not only inherent to the modality itself, but is affect by the spectral bias of neural networks, which can be mitigated through Fourier mappings.

**3D Visual Representations for Imitation Learning.** 3D inputs can be leveraged in different ways, as stand-alone modalities (e.g., point clouds or point maps) or in combination with RGB. On a number of challenging tasks, imitation learning with lightweight point cloud-based policies consistently outperforms RGB and RGB-D modalities while requiring significantly less data (Ze et al., 2024; Zhu et al., 2024). Additionally, Zhu et al. (2024) observe that training on point maps yields no advantage over training on point clouds. RVT (Goyal et al., 2023) re-renders virtual viewpoints as seven-channel point maps containing RGB, depth, and global coordinate information to decouple the current observation from the input used for downstream decision-making, using point clouds only as a intermediate representation. A variety of hybrid 2D/3D approaches (Ke et al., 2025; Gervet et al., 2023; Wilcox et al., 2025) lift 2D features from pre-trained image encoders into 3D space by concatenating them with their 3D positions reconstructed from the original depth maps. This allows models to combine the benefits of pre-trained visual encoders with the explicit 3D representation of point clouds. While these works emphasize architectural design or multi-view fusion, our method focuses on 3D representations, introducing non-parametric Fourier mappings that can be combined with any 3D encoder to make fine geometric details more accessible.

**Deep Learning with Fourier Features.** Fourier features (Tancik et al., 2020; Mildenhall et al., 2021) mitigate the spectral bias of neural networks (Rahaman et al., 2018; 2019), i.e. their tendency to learn low-frequency components faster than high-frequency ones. This bias is amplified by the data manifold geometry where variations that look high-frequency along it may correspond to low-frequency modes in the ambient space (Rahaman et al., 2019), causing fine details to be suppressed. These effects explain why many architectures struggle to condition on fine geometric details, and motivate our use of Fourier feature mappings. Fourier mappings address this by lifting low-dimensional inputs such as Cartesian coordinates into sinusoidal embeddings with multiple frequencies, which can be fixed or learnable (Gao et al., 2023; Sun et al., 2024). Neural radiance fields (Mildenhall et al., 2021; Barron et al., 2022) use this technique to be able to learn detailed 3D scenes with high fidelity, and they are only able to learn blurry, oversmoothed representations without it (Tancik et al., 2020). Adapt3R (Wilcox et al., 2025) propose a novel observation encoder that outperforms other architectures on novel viewpoints unseen during training. While they show the benefit of Fourier features for their architecture, they do not investigate their effect in other contexts. In contrast, we apply Fourier mappings systematically across 3D modalities in diffusion-based IL, providing a frequency-domain perspective that complements architectural approaches.

## 3 METHOD

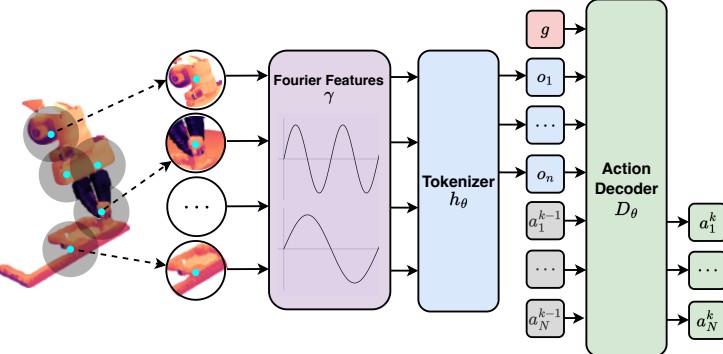

Figure 2: **Framework Overview.** Given a point cloud, we first map each point $i$ and its neighbourhood $\mathcal{N}(i)$ (indicated by the encircled patches) to Fourier feature space. This amplifies subtle geometric differences in each neighborhood. The tokenizer extracts and aggregates features for each neighborhood to produce a set of tokens which are then forwarded to a goal-conditioned diffusion policy $D_\theta$ to denoise the next chunk of actions.

### 3.1 PROBLEM FORMULATION

Imitation Learning aims to learn a policy from expert demonstrations. We are given a dataset containing $N$ expert trajectories $\mathcal{D} = \{\tau_i\}_{i=1}^{N}$, where each trajectory $\tau_i = (\mathbf{g}_i, (o_1, a_1), (o_2, a_2), \ldots, (o_K, a_K))$, where $K$ is the trajectory length and $\mathbf{g}_i$ is the language description for the trajectory. The objective is to learn a policy $\pi(\bar{a}|o, \mathbf{g})$ that maps observations $o$ and embedded goal $\mathbf{g}$ to a sequence of actions $\bar{a} = (a_k, a_{k+1}, \ldots, a_{k+H})$. Predicting sequences of actions, i.e. action chunking, results in more temporally correlated trajectories compared to predicting individual actions (Zhao et al., 2023). Each observation $o$ contains depth images from $M$ cameras. In combination with the camera intrinsic and extrinsic parameters from calibration, we can construct any desired 3D observation representation from these depth images.

### 3.2 SCORE-BASED DIFFUSION

To learn policies from expert demonstrations, we use the typical EDM framework (Karras et al., 2022; Reuss et al., 2023) for score-based action diffusion conditioned on observations of the scene. Diffusion models are generative models that learn to generate new samples through learning to reverse a Gaussian perturbation process. The policy $\pi_\theta(\bar{a}|o)$ is formulated as a score-based diffusion model that can be used to successively denoise actions generated from Gaussian noise back to the data manifold. This perturbation and its inverse process can be expressed with the following Stochastic Differential Equation

$$\mathrm{d}\bar{a} = \left(\beta_t \sigma_t - \dot{\sigma}_t\right)\sigma_t \nabla_a \log p_t(\bar{a}|o, \mathbf{g})dt + \sqrt{2\beta_t}\sigma_t d\omega_t, \tag{1}$$

where $\beta_t$ determines the noise injection rate at diffusion time step $t$, $d\omega_t$ represents infinitesimal Gaussian noise, and $p_t(\bar{a}|o, \mathbf{g})$ denotes the score function of the diffusion process. During policy sampling (i.e. the reverse process), action samples are guided towards high-density regions of the data distribution. To learn this score, we train a neural network $D_\theta$ via score matching (Vincent, 2011):

$$\mathcal{L}_{\mathrm{SM}} = \mathbb{E}_{\sigma, \bar{a}, \boldsymbol{\epsilon}}\left[\alpha(\sigma_t)|D_\theta(\bar{a} + \boldsymbol{\epsilon}, o, \mathbf{g}, \sigma_t) - \bar{a}|_2^2\right], \tag{2}$$

where $D_\theta(\bar{a} + \boldsymbol{\epsilon}, o, \sigma_t)$ represents our trainable neural architecture.

After training, we can generate new action sequences beginning with Gaussian noise by iteratively denoising the action sequence with a numerical ODE solver. Our approach utilizes the DDIM-solver, a specialized numerical ODE-solver for diffusion models (Song et al., 2021) that enables efficient action denoising in a minimal number of steps.

### 3.3 POINT CLOUDS

Given a set of depth images from $M$ cameras $D^{(0)}, \ldots, D^{(M)} \in \mathbb{R}^{W \times H}$ as well as the intrinsic matrices $K_{\mathrm{int}}^{(0)}, \ldots, K_{\mathrm{int}}^{(M)} \in \mathbb{R}^{3 \times 3}$, we first construct point clouds $X^{(j)} \in \mathbb{R}^{WH \times 3}$ in each camera's local coordinate frame via unprojection:

$$X_{iW+j}^{(m)} = (K_{\mathrm{int}}^{(m)})^{-1} \left(i \cdot D_{i,j}^{(m)}, j \cdot D_{i,j}^{(m)}, D_{i,j}^{(m)}\right)^T \tag{3}$$

By multiplying each point cloud with its corresponding extrinsic matrix, we can transform it from the camera coordinate frame to the world frame. The final point cloud $\mathbf{X}$ is obtained by concatenating point clouds from all $M$ views.

We treat point clouds as graphs, where the coordinates XYZ are the node features $\mathbf{x}^0$. This allows us to formulate the point cloud encoder as a message-passing Graph Neural Network (GNN) (Scarselli et al., 2009), a flexible framework that encompasses numerous well-known architectures. Each step $l$ computes new node features

$$\mathbf{x}_i^l = h_\theta^l(\mathbf{x}_i^{l-1}, \bigoplus_{j \in \mathcal{N}(i)} h_\phi^l(\mathbf{x}_i^{l-1}, \mathbf{x}_j^{l-1})), \tag{4}$$

where $\mathcal{N}(i)$ is the neighborhood of point $i$, e.g., $i$'s $k$ nearest neighbors or, in the case of a complete graph, all other nodes. The permutation invariant aggregation $\bigoplus$ can be instantiated as a sum,

max, or mean, and $h_\theta^l$ and $h_\phi^l$ denote learnable parametrized functions. After the final step, the tokenized embedding of the observed point cloud $\{\mathbf{T}_i\} \in \mathbb{R}^{n \times d}$ is a subset of the node features $\mathbf{T}_i = \mathbf{x}_S^L$, where $S = f_{\text{sampling}}(\cdot) \subseteq [n]$ are indices selected by some sampling function. Each token may optionally be augmented with a positional encoding $\mathbf{T}_i \leftarrow \mathbf{T}_i + \text{PE}_\psi(\mathbf{x}_i^0)$ based on the cartesian coordinates of the corresponding point, where $\text{PE}_\psi$ represents some (potentially parametric) function.

### 3.4 FOURIER FEATURE MAPPING

Despite the fact that neural networks are universal function approximators (Hornik et al., 1989), they are biased toward learning low-frequency components first, while high-frequency components converge slowly or may not be learned at all (Rahaman et al., 2019; Tancik et al., 2020). However, an imitation learning policy parametrized by a neural network may need to learn a high frequency function to represent a sharp decision boundary, such as whether to reposition a grasped object or insert it. For a diffusion denoising model, this would allow the network to represent a score function that is a high-frequency function of the scene geometry, though not necessarily of the actions. In 3D point clouds, a Fourier feature mapping allows the network to better distinguish nearby points, which have extremely similar features in Cartesian space.

In contrast to previous work that adds Fourier features to specific, novel architectures (Wilcox et al., 2025), we hypothesize that applying a Fourier feature mapping to Cartesian points feature benefits essentially *any* point cloud-based policy. We adopt a NeRF-style, axis-aligned Fourier feature mapping (Mildenhall et al., 2021). Let $\mathbf{p} = (x, y, z) \in \mathbb{R}^3$ define a Cartesian point. The encoding function $\gamma : \mathbb{R} \to \mathbb{R}^{2L}$ applied separately to the three coordinate values in $\mathbf{p}$ is defined as

$$\gamma_k(x) = \left[ \sin\left(\frac{2\pi x}{\lambda_k}\right), \cos\left(\frac{2\pi x}{\lambda_k}\right) \right]^{\mathbf{T}}, \qquad \lambda_k = \lambda_{\max} \left(\frac{\lambda_{\min}}{\lambda_{\max}}\right)^{\frac{k-1}{L-1}}, \qquad k = 1, \ldots, L. \quad (5)$$

As the Fourier feature mapping is periodic, the point cloud must be bounded by the interval $[-\lambda_{\max}/2, \lambda_{\max}/2]$ to ensure unique features. If this is not possible, the input Cartesian coordinates can be concatenated with the Fourier features, which always yields a unique mapping.

### 3.5 DATA AUGMENTATION

As shown in (Tancik et al., 2020), the choice of wavelengths is essential, as too short wavelengths can cause the network to overfit on the data, while too long wavelengths do not resolve the spectral bias. Instead of carefully tuning the wavelengths to each task, we choose a consistent set of wavelengths and use data augmentation to train the network to ignore frequencies that do not contain useful information. To achieve this, we apply VariableJitter (Gyenes et al., 2025), which avoids the difficulty of tuning the amplitude of typical Gaussian jitter. While uniform jitter applies noise $\epsilon \sim \mathcal{U}(-\sigma, \sigma)$ to each point drawn from a static distribution, VariableJitter samples a new $\sigma \sim \mathcal{U}(0, \sigma_{\max})$ for each point cloud from a uniform distribution. This achieves a trade off between augmenting the data to reduce overfitting and ensuring there is no gap between training and testing data.

## 4 EXPERIMENTS

### 4.1 BENCHMARKS AND DATASETS

We evaluate our approach on two widely used simulation benchmarks, RoboCasa (Nasiriany et al., 2024) and ManiSkill3 (Tao et al., 2025), as well as on three challenging real-world tasks (Jia et al., 2025). Figure 3 provides a visualization of our simulation tasks. All models are trained in a multi-task setting, where the policy is provided a goal description in text form. RoboCasa and real world tasks include two static cameras and an in-hand camera, while ManiSkill3 tasks use only a static camera. In order to highlight the effect of Fourier features on 3D representations, we do not include color features in observations in simulation tasks. The different task groups are shown in Tables 3, 4, and 5 for RoboCasa, ManiSkill3, and the real world, respectively.

**RoboCasa.** RoboCasa (Nasiriany et al., 2024) includes high-precision, long-horizon manipulation tasks in visually rich kitchen scenes. In our study, we focus on 16 tasks that stress fine geometric

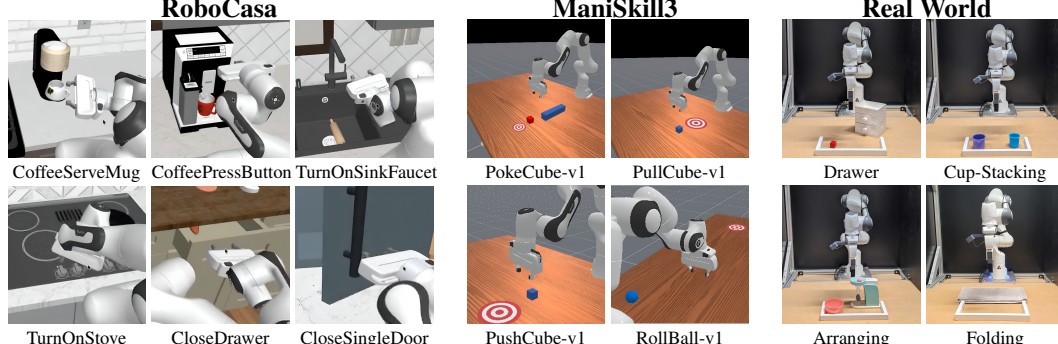

Figure 3: **Overview of all evalution tasks from RoboCasa, ManiSkill3, and Real World bench-marks.** The left column shows a representative subset of six RoboCasa tasks selected from the full set of 16 evaluation tasks. The middle column shows all evaluated ManiSkill3 tasks, and the right column shows all real-world tasks.

alignment and contact, which is where spectral bias is most detrimental. For each task we use 50 human-collected demonstrations provided by RoboCasa. More details on RoboCasa can be found in Appendix A.1.

**ManiSkill3.** We further test on ManiSkill3 (Tao et al., 2025), where we evaluate on four tasks covering grasping and tool usage. Since the majority of tasks use color information to indicate some aspect of the target, we map the target's Cartesian coordinates to Fourier features and pass this as an additional observation token. We train on 500 (RL-generated) demonstrations from each task. Appendix A.1 provides full details on ManiSkill3.

**Real World.** Finally, we adapt four challenging real-world tasks (Jia et al., 2025) that feature long horizons, multiple phases, and precise manipulation. Each task is comprised of distinct sub-tasks with different goal descriptions (e.g. "stack the red cup in the blue cup" for the stack task). For these complex tasks involving color information, we adopt a multi-modal approach utilizing RGB images and point clouds.

Points beyond a maximum depth (2m in ManiSkill3 and the real world and 10m in RoboCasa are removed. In ManiSkill and the real world, point clouds are further cropped to remove the irrelevant background points, and in ManiSkill, the table surface is also removed. We apply voxel downsampling with a voxel size of 0.006 for ManiSkill3 and the real world and 0.01 for RoboCasa. We sample a $\bar{\sigma}_{max}$ for VariableJitter up to 0.002 for simulation environments and 0.001 for the real world, which still allows crucial geometric details to be preserved. Camera observations are resized to $128 \times 128$ for point cloud-based policies and $224 \times 224$ for point map-based policies.

## 4.2 ARCHITECTURES AND BASELINES

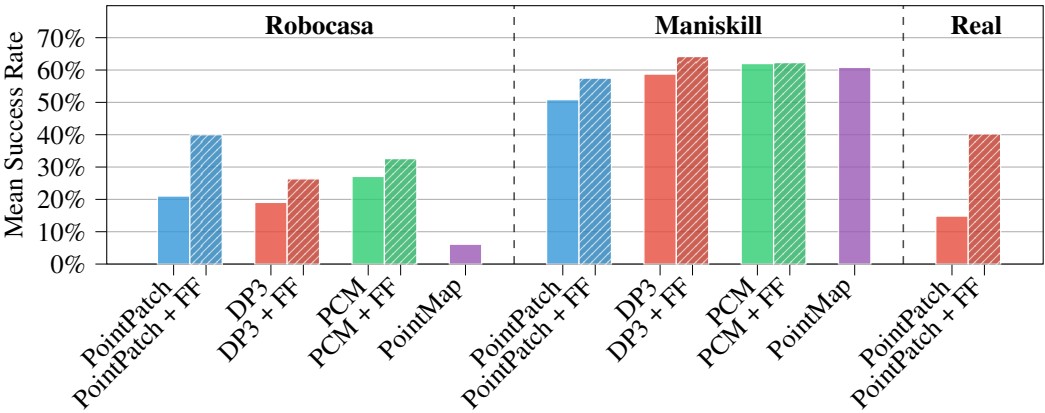

Figure 4: Mean success rate across all tasks of 3D encoders with and without Fourier features on RoboCasa (left) and ManiSkill3 (middle), and the real world (right).

We instantiate the denoising diffusion model as a decoder-only transformer, which receives as input a goal token, a noise level token, a number of observation tokens, and $H$ noisy action tokens, where $H$ is the action horizon. A frozen CLIP RN-50 model (Radford et al., 2021) is used to embed the goal description into a token, while the noise level is encoded as in DDPM (Ho et al., 2020). A learned token position embedding is added onto each token, except for observation tokens when the number of tokens is variable. After the transformer layers, the final $H$ tokens are passed through a linear layer to arrive at denoised actions. We experiment with architectures for encoding point cloud observations into tokens to demonstrate the efficacy of Fourier features for diffusion-based imitation learning for robotic manipulation. The architectures differ in how they implement Equation 4, namely with different aggregations and parameterizations for the learnable functions.

**PointPatch.** The PointPatch encoder (Pang et al., 2022; Yu et al., 2022; Gyenes et al., 2024) divides a given point cloud into overlapping patches, tokenizes each patch and then uses a transformer for token processing. Point features are the Cartesian coordinates relative to the patch center, and are encoded with a lightweight PointNet (Qi et al., 2017b) to create patch tokens. Token position embeddings are computed by passing each centroid position through a two-layer MLP (Pang et al., 2022). Fourier features projections are applied to the relative patch coordinates as well as the centroid positions.

**DP3 Encoder.** Unlike patch-based methods, the DP3 encoder (Ze et al., 2024) creates a single token that embeds information from the entire point cloud. Point features are passed through an MLP, followed by a max-pooling operation to obtain order-invariant global feature. A final projection head maps the embedding to the token dimension, resulting in $\mathbf{T} \in \mathbb{R}^{1 \times D}$. Although this architecture is simple, it is quite data efficient due to its small number of parameters. We do not apply FPS sampling before the encoder since this decreased performance in our experiments.

**PointCloudMatters Encoder.** We additionally evaluate the PointCloudMatters-PointNet (PCM) encoder used in Zhu et al. (2024). This architecture is based on the patching paradigm introduced in PointMAE (Pang et al., 2022), but applies a max aggregation across patches, followed by a final projection head to compute a single token for the entire point cloud. In addition, each point is assigned its absolute Cartesian coordinates as well as its relative coordinates within the patch as features.

**Pointmap Encoder.** To compare against 3D representations that do not follow the encoder blueprint of Equation 4, we also evaluate point maps (Wang et al., 2024; Jia et al., 2025), which contain the same information as point clouds but are arranged in a 2D grid. Given depth images from multiple cameras and their intrinsics and extrinsics parameters, we unproject each pixel into 3D and transform it into the world frame, resulting in a dense point map $\mathbf{X} \in \mathbb{R}^{H \times W \times 3}$ for each camera. The resulting 3D representation can be processed directly with convolutional backbones such as ConvNeXt V2 (Woo et al., 2023) or ResNet (He et al., 2015).

**RGB+PointPatch Encoder.** Due to the flexible transformer design of our denoising model, multi-modal observations can be processed by concatenating tokens from parallel observation encoders. We encode the RGB stream from each camera into a token with a ConvNeXt V2 nano ($\sim$15M parameters) (Woo et al., 2023) that is initialized from a pretrained checkpoint. Point clouds are processed by the PointPatch encoder.

### 4.3 SETUP AND EXPERIMENTS.

**Research Questions.** Our experiments are designed to answer the following research questions: **Q1)** Do Fourier features yield consistent benefits across point cloud encoders and benchmarks? **Q2)** To what extend does the parameterization of these features matter? **Q3)** Does the benefit of Fourier feature mappings translate from simulation to real-world tasks?

To answer these questions, we first explore a series of simulation benchmarks and evaluate several point cloud encoders with and without Fourier features. We then conduct an extensive parameter study. Finally, we train our policies on real-world data and evaluate them on a real robot.

**Training and Evaluation.** Each method is trained for 100 epochs with 3 random seeds, and we test performance after the 60th, 80th, and 100th epochs. We measure the average success rate across 20 rollouts and select the best-performing checkpoint for each seed.

| Category | Task | PointPatch | PointPatch + FF | DP3 | DP3 + FF | PCM | PCM + FF | PointMap |
|---|---|---|---|---|---|---|---|---|
| Insertion | CoffeeServeMug | $0.0_{\pm0.0}$ | $\mathbf{5.0_{\pm8.7}}$ | $3.3_{\pm2.9}$ | $3.3_{\pm2.9}$ | $5.0_{\pm0}$ | $3.3_{\pm2.9}$ | $0.0_{\pm0.0}$ |
| | CoffeeSetupMug | $0.0_{\pm0.0}$ | $\mathbf{3.3_{\pm5.8}}$ | $0.0_{\pm0.0}$ | $0.0_{\pm0.0}$ | $\mathbf{1.7_{\pm2.9}}$ | $0_{\pm0}$ | $0.0_{\pm0.0}$ |
| Pressing Buttons | CoffeePressButton | $18.3_{\pm2.9}$ | $\mathbf{38.3_{\pm5.8}}$ | $18.3_{\pm2.9}$ | $\mathbf{28.3_{\pm10.4}}$ | $21.7_{\pm2.9}$ | $\mathbf{26.67_{\pm2.9}}$ | $8.3_{\pm7.6}$ |
| | TurnOnMicrowave | $10.0_{\pm5.0}$ | $\mathbf{43.3_{\pm23.6}}$ | $26.7_{\pm12.6}$ | $\mathbf{43.3_{\pm12.6}}$ | $25.0_{\pm5.0}$ | $\mathbf{40.0_{\pm5.0}}$ | $0.0_{\pm0.0}$ |
| | TurnOffMicrowave | $15.0_{\pm5.0}$ | $\mathbf{38.3_{\pm5.8}}$ | $28.3_{\pm7.6}$ | $\mathbf{38.3_{\pm2.9}}$ | $37_{\pm14}$ | $\mathbf{51.7_{\pm2.9}}$ | $11.6_{\pm2.9}$ |
| Turning Levers | TurnOnSinkFaucet | $30.0_{\pm13.2}$ | $\mathbf{35.0_{\pm8.7}}$ | $16.7_{\pm7.6}$ | $\mathbf{23.3_{\pm5.8}}$ | $26.7_{\pm7.6}$ | $\mathbf{36.7_{\pm7.6}}$ | $20.0_{\pm10.0}$ |
| | TurnOffSinkFaucet | $41.7_{\pm5.8}$ | $\mathbf{66.7_{\pm2.9}}$ | $43.3_{\pm5.8}$ | $\mathbf{61.7_{\pm12.6}}$ | $61.7_{\pm2.9}$ | $61.7_{\pm5.8}$ | $13.3_{\pm10.4}$ |
| | TurnSinkSpout | $41.7_{\pm5.8}$ | $\mathbf{73.3_{\pm5.7}}$ | $\mathbf{46.7_{\pm2.9}}$ | $41.7_{\pm2.9}$ | $53_{\pm13}$ | $40.0_{\pm5.0}$ | $31.7_{\pm11.5}$ |
| Twisting Knobs | TurnOnStove | $18.3_{\pm10.4}$ | $\mathbf{36.7_{\pm2.9}}$ | $21.7_{\pm5.8}$ | $\mathbf{31.7_{\pm10.4}}$ | $36.7_{\pm7.6}$ | $\mathbf{38.3_{\pm2.9}}$ | $3.3_{\pm2.9}$ |
| | TurnOffStove | $5.0_{\pm0.0}$ | $\mathbf{11.7_{\pm2.9}}$ | $16.7_{\pm5.8}$ | $16.7_{\pm7.6}$ | $\mathbf{21.7_{\pm7.6}}$ | $18.3_{\pm2.9}$ | $6.7_{\pm2.9}$ |
| Open/Close Drawers | OpenDrawer | $3.3_{\pm2.9}$ | $\mathbf{18.3_{\pm2.9}}$ | $3.3_{\pm2.9}$ | $\mathbf{10.0_{\pm5.0}}$ | $6.7_{\pm2.9}$ | $\mathbf{20_{\pm5}}$ | $0.0_{\pm0.0}$ |
| | CloseDrawer | $33.3_{\pm7.6}$ | $\mathbf{70.0_{\pm0.0}}$ | $40.0_{\pm17.3}$ | $\mathbf{53.3_{\pm7.6}}$ | $60_{\pm13}$ | $60_{\pm17}$ | $3.3_{\pm2.9}$ |
| Open/Close Doors | OpenSingleDoor | $21.7_{\pm2.9}$ | $\mathbf{36.7_{\pm7.6}}$ | $6.7_{\pm5.8}$ | $\mathbf{11.7_{\pm5.8}}$ | $10.5_{\pm5.0}$ | $\mathbf{16.7_{\pm2.9}}$ | $0.0_{\pm0.0}$ |
| | CloseSingleDoor | $\mathbf{76.7_{\pm7.6}}$ | $63.3_{\pm5.8}$ | $23.3_{\pm2.9}$ | $\mathbf{41.7_{\pm7.6}}$ | $38_{\pm12}$ | $\mathbf{48.3_{\pm7.6}}$ | $0.0_{\pm0.0}$ |
| | OpenDoubleDoor | $11.7_{\pm5.8}$ | $\mathbf{28_{\pm10}}$ | $0.0_{\pm0.0}$ | $\mathbf{1.6_{\pm2.9}}$ | $5.0_{\pm5.0}$ | $\mathbf{10.0_{\pm5.0}}$ | $0.0_{\pm0.0}$ |
| | CloseDoubleDoor | $23.3_{\pm5.8}$ | $\mathbf{63.3_{\pm5.8}}$ | $10.0_{\pm5.0}$ | $\mathbf{15.0_{\pm0.0}}$ | $23.3_{\pm5.8}$ | $\mathbf{50_{\pm10}}$ | $0.0_{\pm0.0}$ |
| **Average Success Rate** | | $21.9$ | $\mathbf{39.5}$ | $19.1$ | $\mathbf{26.3}$ | $27.1$ | $\mathbf{32.6}$ | $6.1$ |

Table 1: Average success rates on different Robocasa tasks across 3 seeds. Fourier features generally lead to significant improvements for both PointPatch, DP3, and PCM architectures. In contrast, the convolutional PointMap struggles on these tasks, likely due to task complexity and data sparsity.

**Fourier Features.** We use a fixed set of $L{=}8$ Fourier bands with log-spaced wavelengths between $\lambda_{\max}{=}4.0$ and $\lambda_{\min}{=}0.06$ for all experiments. The choice of $\lambda_{\max}$ ensures both the sin and cos components of the largest band are unique within the task space, which is typical bounded to roughly $[-1, 1]^3$, while $\lambda_{\min}$ is small enough to discriminate neighboring voxels. The resulting $D{=}3{\times}(2L){=}48$ Fourier features per Cartesian point encode positions across this scale range, ranging from a global encoding at $\lambda_{\max}$ to a voxel-level encoding at $\lambda_{\min}$.

## 5 RESULTS

**Quantitative Results.** Figure 4 shows the average success rate over all tasks for RoboCasa and ManiSkill3, while Tables 1 and 2 provide detailed results for both benchmarks, respectively. Across both benchmarks, we observe that Fourier feature mappings significantly boost the success rate in the aggregate and on a large majority of individual tasks. In RoboCasa, success rates on individual tasks jump by as much as 35%. For example, PointPatch on CloseDrawer improves from 33.3% to 70.0%, and CloseDoubleDoor improves from 23.3% to 63.3%, while the overall average increases from 21.9% to 39.5%. The results suggest that Fourier mappings help preventing the spectral bias and expose high-frequency geometric cues, making them a robust input representation across a variety of tasks. Further, these improvements are not tied to a specific architecture and appear for tasks of any difficulty. For example, the OpenDrawer task improves from almost no success at all to 18.3%. Despite using substantially different architectures, DP3 and PCM show similar trends. While DP3 starts with a lower base performance, Fourier features still result in significant improvements for 12 of 16 tasks in RoboCasa and 3 of 4 tasks in ManiSkill, while achieving at least parity in every single case. PCM similarly improves in 11 of 16 tasks in RoboCasa, but does not significantly improve on the easier ManiSkill tasks. In comparison, point maps do not perform competitively on RoboCasa, and slightly underperform DP3 and PCM on the simpler ManiSkill tasks. This result may indicate a lower data efficiency of point maps, since we train on 500 demonstrations for each ManiSkill task and only 50 for each RoboCasa task, and is consistent with prior work (Zhu et al., 2024) which shows that simple point cloud encoders outperform point maps in low data regimes.

We evaluate our best-performing method, PointPatch, with and without Fourier features for 16 rollouts on each real world task. Our setup consists of two static Zed Mini cameras and a RealSense D405 as an in-hand camera (see Figure 5). Policies with Fourier features achieved significantly higher scores on all tasks, with an aggregate improvement from 14.8% to 40.23%. Notably, the policy without Fourier features was unable to pick up the cup in the Cup-Stacking task a single time, often because it knocked the cup over while trying to grasp it. The least benefit was observed on the Folding task, which also requires the least geometric precision to carry out. These results demon-

| Tasks | Demos Per Task | Methods with Scores | | Max |
|---|---|---|---|---|
| | | RGB+ PointPatch | RGB+ PointPatch+FF | |
| Drawer | 102 | 0.3125 | **1.625** | 4 |
| Cup-Stacking | 80 | 0 | **0.625** | 2 |
| Arranging | 100 | 0.3125 | **1.3125** | 4 |
| Folding | 75 | 1.3125 | **1.6875** | 3 |

Figure 5: Real-world drawer experiments. Left: task setup. Middle: RGB views from three cameras and depth view from left camera. Right: average scores across 16 rollouts on three real-world high-precision tasks.

strate the Fourier features are robust to real world conditions such as noisy depth measurements, occlusion artifacts, and camera miscalibration.

**Qualitative Results.** Qualitatively, we also notice that policies trained on Fourier feature mappings move faster and more decisively, and more closely imitate the demonstration data. Policies trained without Fourier features tend to hesitate before making contact with objects, or behave as if they cannot perceive the scene. In the real world, the policies without Fourier features were often catastrophically unable to learn the data.[1] We hypothesis that due to the interaction of spectral bias with the max operation, these policies were overfitting on some global features of the observed point cloud rather than responding to the fine geometry near the end effector.

| Category | Task | PointPatch | PointPatch + FF | DP3 | DP3 + FF | PCM | PCM + FF | PointMap |
|---|---|---|---|---|---|---|---|---|
| Table-Top 2 Finger Gripper | PullCube-v1 | $56.7_{\pm7.6}$ | $63.3_{\pm5.8}$ | $91.7_{\pm5.8}$ | $80.0_{\pm13.2}$ | $77_{\pm10}$ | $73.3_{\pm3.0}$ | $63.3_{\pm7.6}$ |
| | PushCube-v1 | $68.3_{\pm7.6}$ | $75.0_{\pm5.0}$ | $51.7_{\pm16.1}$ | $76.7_{\pm16.1}$ | $74.0_{\pm2.0}$ | $76.7_{\pm4.2}$ | $78.3_{\pm7.6}$ |
| | PokeCube-v1 | $56.7_{\pm12.6}$ | $68.3_{\pm5.8}$ | $61.7_{\pm2.9}$ | $63.3_{\pm2.9}$ | $71.3_{\pm5.0}$ | $68.0_{\pm4.0}$ | $71.7_{\pm10.4}$ |
| | RollBall-v1 | $21.7_{\pm7.6}$ | $23.3_{\pm7.6}$ | $30.0_{\pm13.2}$ | $36.7_{\pm2.9}$ | $25.3_{\pm1.2}$ | $31_{\pm11}$ | $30.0_{\pm5.0}$ |
| **Average Success Rate** | | 50.8 | **57.5** | 58.8 | **64.2** | 62.0 | **62.3** | 60.8 |

Table 2: Average success rates on Maniskill tasks across 3 seeds. Fourier features improve the performance of point-cloud based architectures, likely because they enable better differentiation of fine-grained details. For Maniskill, PointMaps are competitive with approaches enhanced with Fourier features, presumably due to larger training datasets.

# 6 ANALYSIS

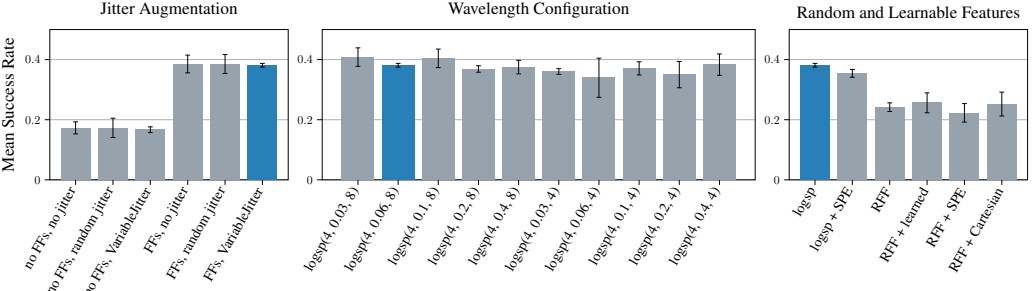

Figure 6: Parameter study on Fourier features and data augmentation. Our default parameters for other experiments are marked in blue. **Left**: While jitter does not affect average performance, VariableJitter reduces variance both with and without Fourier features. **Middle**: Performance is robust to different configurations of log-spaced wavelengths, denoted as $\mathrm{logsp}(\lambda_{max}, \lambda_{min}, L)$. **Right**: Log-spaced, axis-aligned frequencies perform better than sampling randomly from a Gaussian (RFF). Learning the frequencies either directly or with SPE does not show a consistent benefit.

---

[1]Rollout videos on real world tasks at our project page: https://fourier-il.github.io/fourier-il.

For additional experiments, we use a reduced set of 8 RoboCasa tasks (Pressing Buttons, Turning Levers, and Twisting Knobs task groups) and train a single policy on all 8 tasks. We train 3 seeds for each method and report the mean and standard deviation across seeds.

**Jitter** To investigate the robustness of Fourier features, we consider different options for the jitter data augmentation, namely no jitter, standard random jitter drawn from $\sim \mathcal{U}(-\sigma_{\max}, \sigma_{\max})$, and VariableJitter as described in Subsection 3.5. Figure 6 (**left**) shows that adding jitter does not make a large difference in terms of average performance. However, VariableJitter greatly reduces the variance of the results both with and without Fourier features, likely because it acts as data augmentation to make the learned policies more robust and reliable.

**Wavelengths.** We experiment with different numbers of wavelengths and a range of minimum wavelengths in Figure 6 (**middle**). While Fourier feature encoding benefits slightly from using 8 instead of 4 wavelengths, it is robust to a wide range of minimum wavelengths. Unlike in related work using Fourier features for NeRFs, we do not see any benefit to task-specific tuning of wavelength ranges. Together with the above, this consistency indicates that Fourier features provide a robust basis for learning challenging robotic manipulation task, and are relatively insensitive to concrete hyperparameters or changes in the training setup.

**Learned and Gaussian Fourier Features.** Sun et al. (2024) find that optimizing the frequencies directly through gradient descent is suboptimal, and they instead propose SPE, which applies a linear layer and a sinusoidal non-linearity instead. Furthermore, instead of purely on-axis frequencies, Tancik et al. (2020) suggest randomly sampling frequency vectors $\mathbf{v} \sim \mathcal{N}(0, \sigma^2)$ from an isotropic normal distribution, where we set $\sigma=10$. In this case, the encoding function $\gamma : \mathbb{R}^3 \to \mathbb{R}^2$ applied to Cartesian point $\mathbf{p}$ is defined as $\gamma_k(\mathbf{p}) = [\sin(2\pi\mathbf{v}_k \cdot \mathbf{p}), \cos(2\pi\mathbf{v}_k \cdot \mathbf{p})]^{\mathbf{T}}$ for $k = 1, \ldots, L$. These Gaussian RFFs can likewise be learned either by directly optimizing frequencies or with SPE. Figure 6 (**right**) evaluates different combinations of log-spaced or Gaussian random frequencies with different methods for learning frequencies. We find that simply using a fixed encoding with log-spaced frequencies works best.

**Frequency Response.** We analyze the FFT spectra of several components of the trained policies to see if architectural changes are reflected in the policy's sensitivity to spatial frequencies. Figure 10 visualizes the spectral responses of policies trained with and without Fourier features, as well as log-spaced, Gaussian random, and learned Fourier features. We find that adding Fourier features measurably increases the frequency response at the small wavelengths relevant for the local patch embedding. While Gaussian RFFs also achieve this, they demonstrate much lower sensitivity to large wavelengths, which is necessary for the patch position encoder to distinguish the global features of the scene. Gaussian RFFs tend to cause the patch position encoder overfit on small wavelengths. In contrast, fixed, log-spaced Fourier features allow the local patch encoder to respond to small wavelengths and the global patch position encoder to focus on large wavelengths.

# 7 CONCLUSION

Neural networks are biased towards learning low-frequency functions of their inputs, while ignoring the high-frequency information that is essential for high-precision manipulation, such as insertion tasks or grasping. We propose to apply the well-known Fourier feature mapping introduced in NeRF (Mildenhall et al., 2021) to a variety of point cloud-based imitation learning methods and test them on high-precision manipulation tasks in simulation and on a real robot.

We demonstrate that, for point cloud architectures, simply encoding the policy's coordinate inputs via Fourier features provides significant and consistent performance benefits. These benefits hold across RoboCasa and ManiSkill3 tasks of varying difficulty, and are consistent across different encoders. On ManiSkill3 tasks, this modification brings point clouds on par with alternative 3D representations such as point maps, while on RoboCasa, they far exceed them. Parameter studies further show that Fourier features are robust to most hyperparameters, making them easy to use at essentially no additional cost. We thus argue that Fourier features should be used almost any point cloud encoder architecture rather than Cartesian point features. Future work may investigate gradient-based learning of the optimal wavelengths or additional regularization to improve scalability.

ETHICS STATEMENT

This work introduces Fourier feature projections to enhance 3D modalities for high-precision imitation learning. While our evaluation emphasizes robotic manipulation, the approach is broadly relevant to other domains where spatial reasoning is required. As with many advances in robot learning, the outcomes depend on the context of deployment: increased accuracy and robustness can yield positive impacts in assistive and industrial settings, but also raise risks if applied irresponsibly. We emphasize that the governance of powerful robotic technologies must extend beyond the research community and individual organizations, requiring oversight by public institutions and democratic processes.

REPRODUCIBILITY STATEMENT

To ensure the reproducibility of our research, we provide a detailed description of our experiments and implementation in Chapter 4, and include a list of hyperparameters in Appendix A.2. The used RoboCasa and ManiSkill datasets are the public available versions which can be found on their corresponding websites. Further information on the datasets used in our experiments can be found in Chapter 4 and in Appendix A.1. Our source code will be released with the final version of the paper.

ON LLM USAGE

Large language models were employed to refine individual phrases during the writing of the paper, to assist with literature search and exploration, and to aid in code implementation. All outputs from large language models were checked verified by the authors at every stage of the project, including text, literature, and code. We also used them in limited ways for generating illustrative visualizations, but used our own images and material as the basis for these visualizations.

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

## A  APPENDIX

### A.1  EXPERIMENT DETAILS

Figure 7: **Overview of RoboCasa Simulation Environments.** Example kitchen scenes and tasks illustrating the diversity of household manipulation settings provided by RoboCasa.

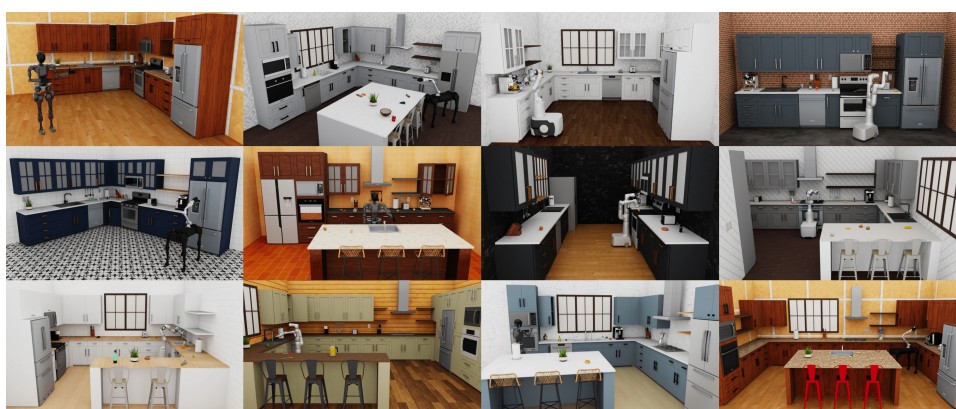

**RoboCasa.**  RoboCasa (Nasiriany et al., 2024) is a large-scale simulation benchmark designed for training generalist robots in realistic household settings, with an emphasis on kitchen environments. It provides 100 tasks in total: 25 atomic tasks with 50 human demonstrations each, and 75 composite tasks with automatically generated demonstrations. The task set covers eight fundamental skills that are essential for home robotics: (1) pick-and-place, (2) door opening and closing, (3) drawer opening and closing, (4) knob turning, (5) lever manipulation, (6) button pressing, (7) insertion, and (8) navigation. To evaluate our method, we selected 16 tasks from the atomic tasks described in Table 3, each representing a different skill. The joint action space is 7-dimensional, including end-effector translation, rotation, and gripper control.

Table 3: RoboCasa evaluation tasks.

| Category | Task | Description |
|---|---|---|
| Insertion | CoffeeServeMug | Remove the mug from the holder and place it on the counter. |
| | CoffeeSetupMug | Place the mug into the coffee machine's mug holder. |
| Pressing Buttons | CoffeePressButton | Press the button to pour coffee into the mug. |
| | TurnOnMicrowave | Start the microwave by pressing the start button. |
| | TurnOffMicrowave | Stop the microwave by pressing the stop button. |
| Turning Levers | TurnOnSinkFaucet | Turn on the sink faucet to start water flow. |
| | TurnOffSinkFaucet | Turn off the sink faucet to stop water flow. |
| | TurnSinkSpout | Rotate the sink spout. |
| Twisting Knobs | TurnOnStove | Turn on a specific stove burner by twisting its knob. |
| | TurnOffStove | Turn off a specific stove burner by twisting its knob. |
| Open/Close Drawers | OpenDrawer | Open a drawer. |
| | CloseDrawer | Close a drawer. |
| Opening and Closing Doors | OpenSingleDoor | Open a microwave door or a cabinet with a single door. |
| | CloseSingleDoor | Close a microwave door or a cabinet with a single door. |
| | OpenDoubleDoor | Open a cabinet with two opposite-facing doors. |
| | CloseDoubleDoor | Close a cabinet with two opposite-facing doors. |

**ManiSkill3.**  ManiSkill3 (Tao et al., 2025) is a large-scale GPU-parallelized simulation benchmark designed for scalable training of embodied agents. It offers diverse object-centric manipulation tasks such as grasping, assembling, and tool use, with support for both imitation and reinforcement

learning. Unlike RoboCasa, which emphasizes long-horizon household tasks in visually rich kitchen environments, ManiSkill3 provides highly parallelized simulation and rendering of physics-based interactions, enabling efficient large-scale experimentation and evaluation of manipulation policies.

A summary of all ManiSkill3 tasks can be found in Table 4, each representing a distinct skill.

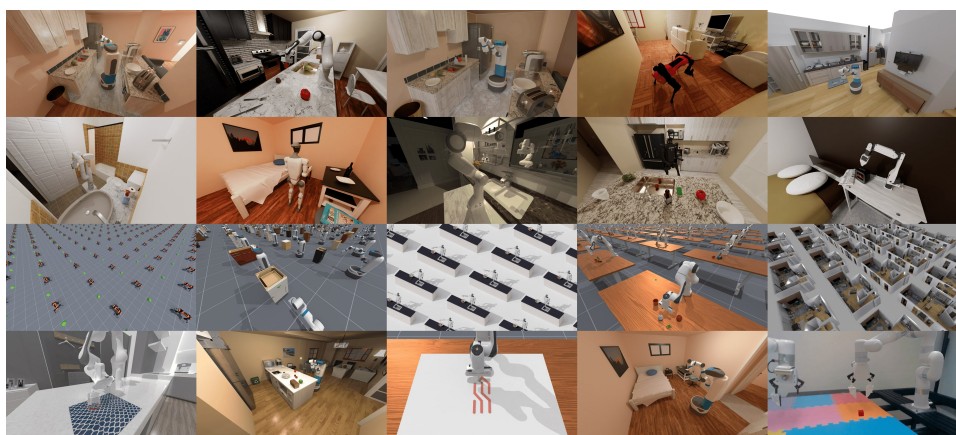

Figure 8: **Overview of ManiSkill3 Simulation Environments.** Example object-centric manipulation tasks illustrating the diversity of interactions supported by ManiSkill3.

Table 4: ManiSkill3 evaluation tasks.

| Category | Task | Description |
|---|---|---|
| Table-Top 2 Finger Gripper | PullCube-v1 | Pick up the cube and pull it to the target. |
| | PushCube-v1 | Push the cube into the target. |
| | PokeCube-v1 | Use the tool to poke the cube until it reaches the target. |
| | RollBall-v1 | Push the ball to make it roll into the target. |

**Real World.** A summary of the real world tasks along with their various goal descriptions can be found in Table 5.

Table 5: Real world tasks with their goal descriptions.

| Task | Descriptions |
|---|---|
| Drawer | put the red cube in the top drawer |
| | put the red cube in the bottom drawer |
| Cup-Stacking | stack the blue cup in the purple cup |
| | stack the orange cup in the red cup |
| | stack the red cup in the blue cup |
| | stack the yellow cup in the orange cup |
| Arranging | put the blue cup in the coffee machine |
| | put the orange cup in the coffee machine |
| | put the pink bowl in the coffee machine |
| | put the red cup in the coffee machine |

## A.2 HYPERPARAMETERS

Table 6: Summary of the Hyperparameters for all of our experiments.

| Hyperparameter | ManiSkill | RoboCasa |
|---|---|---|
| Number of Attention Blocks | 4 | 4 |
| Attention Heads | 4 | 4 |
| Action Chunk Size | 10 | 20 |
| History Length | 1 | 1 |
| Embedding Dimension | 256 | 256 |
| Goal Lang Encoder | CLIP Resnet-50 | CLIP Resnet-50 |
| Attention Dropout | 0.3 | 0.3 |
| Residual Dropout | 0.1 | 0.1 |
| MLP Dropout | 0.1 | 0.1 |
| Optimizer | AdamW | AdamW |
| Betas | [0.9, 0.9] | [0.9, 0.9] |
| Learning Rate | 1e-4 | 1e-4 |
| Weight Decay | 0.05 | 0.05 |
| $\sigma_{\max}$ | 80 | 80 |
| $\sigma_{\min}$ | 0.001 | 0.001 |
| $\sigma_t$ | 0.5 | 0.5 |
| EMA decay | 0.995 | 0.995 |
| Time steps | Exponential | Exponential |
| Sampler | DDIM | DDIM |
| Denoising Steps | 10 | 10 |

**Time**
$\longrightarrow$

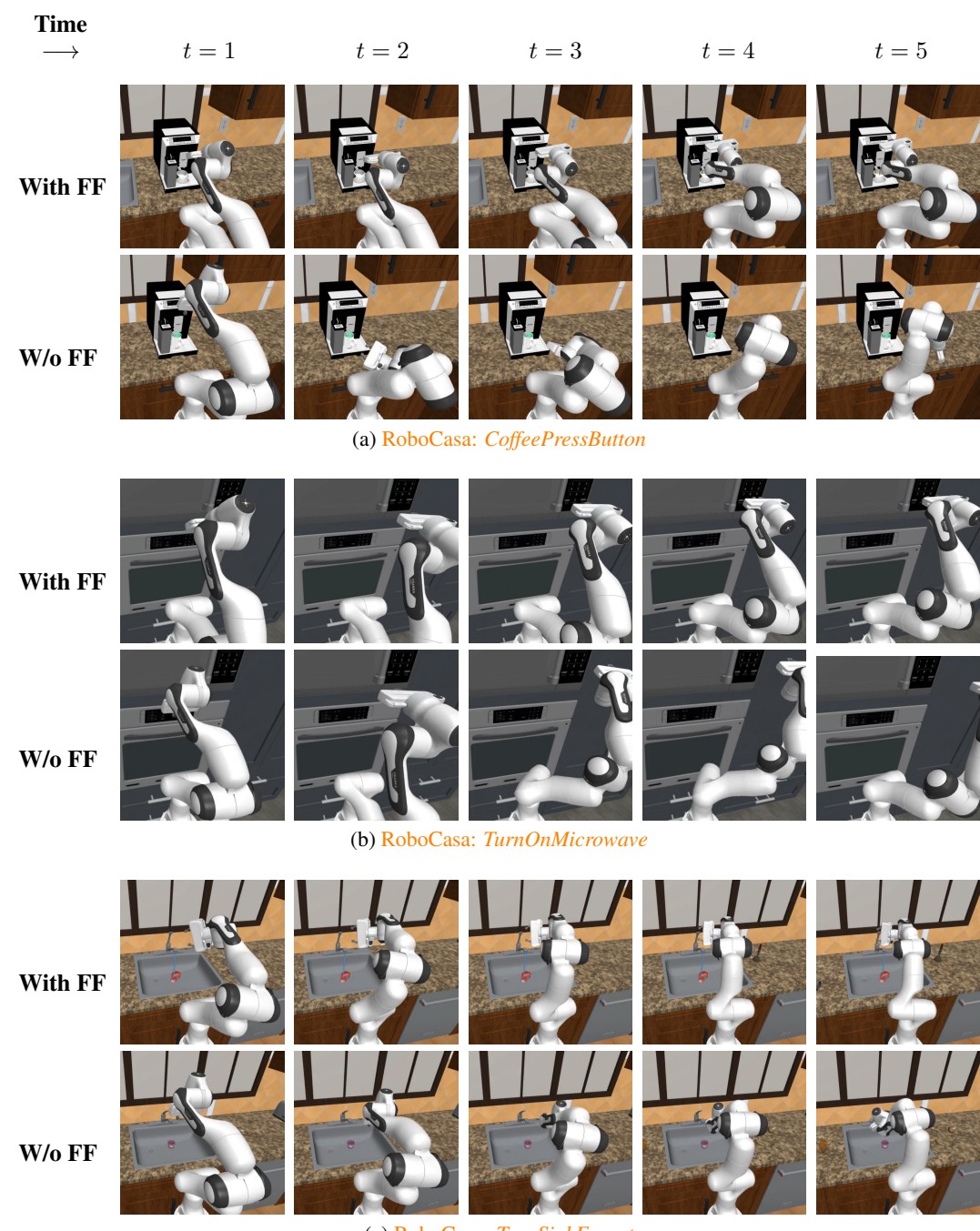

(a) RoboCasa: *CoffeePressButton*

(b) RoboCasa: *TurnOnMicrowave*

(c) RoboCasa: *TurnSinkFaucet*

Figure 9: Qualitative comparison of PointPatch+FF (upper row) and PointPatch (lower row) policies on three RoboCasa tasks. Policies trained without Fourier features have difficulty learning the demonstration data and carrying out complex movements with precision. Time proceeds from left to right in each row.

## A.3 QUALITATIVE RESULTS

Figure 9 shows representative rollouts from PointPatch+FF policies on selected RoboCasa tasks. Overall, the agents trained with Fourier features reliably make contact with the target objects (e.g., buttons and lever handles) and completes all three tasks, whereas the agents trained without Fourier features fail to accomplish the tasks.

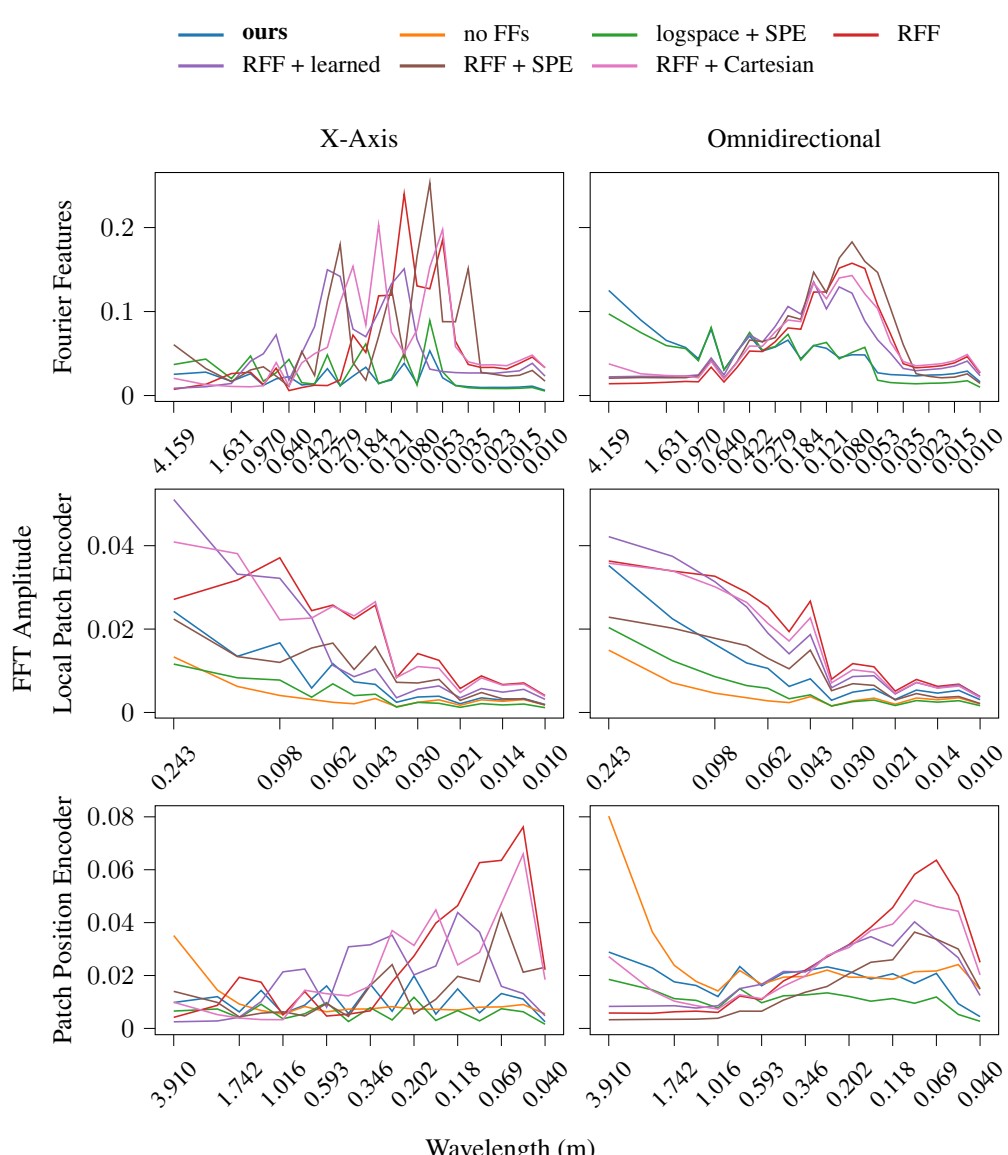

Figure 10: FFT spectra of selected components of policies trained on RoboCasa. FFT amplitudes are averaged across latent features and across random directions for the omnidirectional case. The **top row** shows the FFT spectra from the Fourier feature projection, which may have learnable parameters in the `SPE` and `learned` cases. The Local Patch Encoder **(middle row)** only sees coordinates relative to the patch center, and is therefore exposed only to smaller wavelengths during training. The Patch Position Encoder assigns a position embedding to each token based on the patch's global coordinates, and must therefore be sensitive to larger wavelengths to encode global structure.

## A.4 SPECTRAL ANALYSIS

To investigate how these architectural choices affect the spectral bias of the resulting policies, we probe the sensitivity of selected components of trained policies to a range of wavelengths. We generate query points are regular intervals along the x-axis or along 256 random rays for the omnidirectional case. We compute the FFTs of each latent feature's spectral response and average across directions in the omnidirectional case. FFT amplitudes are further averaged across latent features and re-binned to alleviate the relative density of smaller wavelengths compared to larger ones.

Figure 10 shows that adding Fourier features measurably increases the frequency response at the small wavelengths relevant for the local patch embedding. While Gaussian RFFs also achieve this, they demonstrate much lower sensitivity to large wavelengths, which is necessary for the patch position encoder to distinguish the global features of the scene. Gaussian RFFs tend to cause the patch position encoder overfit on small wavelengths. In contrast, fixed, log-spaced Fourier features allow the local patch encoder to respond to small wavelengths and the global patch position encoder to focus on large wavelengths. This is reflected in the higher success rates of policies trained with Fourier features of this type.

