# OpenReview forum: "Fourier Features Let Agents Learn High Precision Policies with Imitation Learning"
_ICLR.cc/2026/Conference — Submitted to ICLR 2026_

### Official Review · Reviewer_ApuY · 2025-11-01

**Soundness:** 3
**Presentation:** 3
**Contribution:** 3
**Rating:** 6
**Confidence:** 4

**Summary:**

The paper studies spectral bias in point-cloud–conditioned imitation learning (IL) policies and proposes a simple, architecture-agnostic fix: apply a NeRF-style Fourier feature mapping to Cartesian 3D inputs before point-cloud encoding. The authors instantiate this on two representative encoders—PointPatch and DP3—and evaluate on RoboCasa (16 high-precision kitchen tasks, 50 demos each) and ManiSkill3 (4 tabletop tasks, 500 demos each). They report consistent gains from the Fourier mapping: e.g., RoboCasa mean success improves from 21.0%→40.0% (PointPatch) and 19.1%→26.3% (DP3), with notable per-task jumps like CloseDrawer 33.3%→70.0%. On ManiSkill3, average success rises from 50.8%→57.5% (PointPatch) and 58.8%→64.2% (DP3). The approach is intentionally minimal (fixed log-spaced bands; variable-magnitude jitter augmentation) and claims broad applicability to 3D-based IL.

**Strengths:**

1.Clear, simple idea with broad compatibility. The paper targets a real pain point—networks’ low-pass bias on slowly varying XYZ—and plugs in a standard Fourier mapping that can sit in front of most point-cloud tokenizers, not just a bespoke architecture.

2.Solid experimental coverage. Two encoders (local patch tokens vs. global DP3 token) and two popular benchmarks (RoboCasa, ManiSkill3) under a multi-task IL setup; consistent benefits across most tasks and encoders, with visual qualitative evidence.

3.High-precision tasks emphasized. The study focuses on tasks where small geometric distinctions matter (insertions, buttons, levers), which is where spectral bias plausibly bites most; the per-task tables quantify where improvements are largest.

**Weaknesses:**

1.Limited novelty. The core technique (Fourier features / positional encodings) is well-established; the main contribution is a systematic application and study in point-cloud IL.

2.Real-robot validation absent. Claims emphasize high-precision manipulation, but results are purely in simulation (RoboCasa/ManiSkill3). The paper lacks real-world experiments. I would like to see solid real-world experiments to support your claim.

**Questions:**

1.Coordinate bounding & periodicity. You note the mapping is periodic and requires points to lie in [−λmax/2, λmax/2]. How are coordinates normalized/cropped in multi-view reconstruction, and what happens to points outside bounds during exploration or camera drift?

2.Sensitivity to frequency design. How sensitive are gains to L, λmin, λmax? Could learned Gaussian RFF or learned sinusoidal frequencies outperform fixed log-spaced bands here? Please include a small sweep or a learned-RFF variant.

3.Why DP3 sometimes drops. On ManiSkill3 PullCube, DP3 + FF underperforms vanilla DP3 (91.7→80.0). What failure mode explains this, and can frequency ranges be task-adapted to mitigate regressions?

4.How does the method handle depth noise, extrinsic/intrinsic miscalibration, or partial occlusion? I would like to see solid real-world experiments to support your claim.

---

> ### Author Response · Authors · 2025-11-20
>
> We thank the reviewer for their detailed review and for summarizing the intended message of the paper: a simple architecture-agnostic solution to the spectral bias that hinders point cloud-based imitation learning. They also highlighted our extensive simulation experiments with representative encoder architectures, showing meaningful and consistent improvements when using Fourier features. We look forward to engaging with the reviewer in the coming weeks to improve the paper even further.
>
> From their comments, the reviewer’s main concerns are the lack of validation in real-robot experiments, especially in the presence of depth noise and imperfect camera calibration; missing comparison against learned Fourier features or Gaussian random Fourier features (RFFs); and a discussion of task-dependent tuning of frequency ranges. We absolutely agree with the reviewer that these are important points and have started experiments to address them. We will update the draft as soon as we have analyzed the results.
>
> Here, we provide responses to the reviewer’s stated questions:
>
> 1. Indeed, Fourier feature embeddings are periodic, which may cause ambiguities for points lying outside of $[−\lambda_\text{max}/2, \lambda_\text{max}/2]$. In our experiments, points are transformed into the world frame. In ManiSkill, points beyond the edge of the scene ([-2m, 2m] in both x and y) are cropped out of the observation. As a result, we can guarantee that all input coordinates are bounded. In real-world mobile manipulation, it may be more natural to express observations in the robot’s frame of reference, which still allows points in the distance to be cropped away. In cases where boundedness cannot be guaranteed, it is trivial to concatenate the Cartesian coordinates onto the Fourier embedding of each point, which resolves any ambiguities.
> 2. We have started additional experiments with different frequency configurations, including learned sinusoidal frequencies as proposed by Sun et al.[1]. We will update the manuscript with the results.
> 3. Our aim in this paper is to show that a single set of hyperparameters improves success across all tasks and architectures, as this would provide the greatest benefit for the research community. Therefore, we do no task-specific tuning, which naturally results in some outlier results on individual tasks. Nonetheless, adding Fourier features improves performance on 18/20 tasks with PointPatch and 15/20 tasks with DP3. This remarkable level of consistency suggests that the result is robust across tasks an architectures. Ultimately, it is certainly possible to tune the frequencies for each task if desired.
> 4. We agree that real world experiments are essential for validating the robustness of the results. We have started further experiments on real-world robot tasks to investigate this point further.
>
> [1] Sun, Chuanhao, et al. "Learning High-Frequency Functions Made Easy with Sinusoidal Positional Encoding." *International Conference on Machine Learning*. PMLR, 2024.

---

> > ### Author Response · Authors · 2025-11-28
> >
> > We thank the reviewer again for their kind review and for their patience in waiting for the revision. We believe the draft we just uploaded addresses many of the reviewer’s concerns.
> >
> > - We have added extensive real robot experiments on 3 challenging real world tasks, featuring long horizons, multiple phases, and precise manipulation. Even more than our simulation results, these real world results show that Fourier features are robust to real world noise and camera misalignment, improving the average normalized score from 5% to 35%. We direct the reviewer to chapter 5, as well as to our [project page](https://fourier-il.github.io/fourier-il/) for rollout videos.
> > - Extensive ablations of frequency design, which show that our method is quite robust to different settings of L, λmin, λmax. We additionally experiment with learned Gaussian RFFs and learned sinusoidal frequencies. Surprisingly, we find that learning the frequencies provides no significant benefit, while Gaussian random initialization reduces performance. Overall, we find that a simple setup with fixed, logspace frequencies performs best.
> > - We have added a novel encoder architecture, PCM, where we again show a clear gain in success rate when adding Fourier features. As with PointPatch and DP3 encoders, there are outlier tasks, but PCM is improved on 11 of 16 RoboCasa tasks through the use of Fourier features, and the average success rate jumps from 27% to 33%. This result further highlights the consistent benefits of Fourier features for many point cloud encoders.
> >
> > We invite the reviewer to continue to engage in discussion. We look forward to further comments and criticisms, as they ultimately help to improve the quality of the paper.

---

### Official Review · Reviewer_TuVu · 2025-11-01

**Soundness:** 3
**Presentation:** 4
**Contribution:** 2
**Rating:** 4
**Confidence:** 4

**Summary:**

This paper proposes encoding 3D point-cloud positions with Fourier features to help an imitation-learning policy focus on geometric details. The authors conduct experiments on RoboCasa and ManiSkill3 to demonstrate effectiveness. However, the paper still lacks real-world experiments and clear, significant contributions.

**Strengths:**

* This paper highlights a trick that many robotics papers overlook.
* The paper is well presented and easy to read.

**Weaknesses:**

* The novelty is limited. While applying Fourier features is a reasonable addition to the imitation-learning network, this is an incremental contribution, and its impact is difficult to validate without large-scale real-world experiments.
* No real-world experiments. The authors evaluate only on RoboCasa and ManiSkill3, which are known for simplified physics and susceptibility to overfitting. Without convincing real-world results, it is hard to accept this as a substantial contribution to the robotics community.
* The comparison in Fig. 5 is not fair. The proposed policy is compared qualitatively to a baseline, but the two scenarios differ, making it difficult to conclude that the proposed method attends better to details.
* The evaluation appears too noisy to support strong conclusions. For example, in RoboCasa (Fig. 4), PP+FF outperforms DP3+FF, while in ManiSkill3 the opposite holds. This suggests the current benchmarking is not informative enough to determine whether the trick consistently helps.

**Questions:**

* Why was EDM chosen as the action-conditioned diffusion framework instead of the more commonly used DDPM? Are there specific considerations driving this choice?
* According to iCT ([https://arxiv.org/abs/2310.14189](https://arxiv.org/abs/2310.14189)), Fourier features are sensitive to hyperparameters. How were the hyperparameters selected here? The current choices appear somewhat arbitrary.

---

> ### Author Response · Authors · 2025-11-20
>
> The authors extend their sincere thanks to the reviewer for their valuable feedback and thought-provoking questions. This input is essential for the scientific process and we look forward to further discussion during the rebuttal period.
>
> As we understand it, the reviewer’s main concerns are the following: a lack of real-world experiments; the need for more evaluations in simulation, potentially with different architectures, to make the claimed benefit more convincing; and an unfair qualitative comparison in Figure 5. We have started experiments to address these points. We would like to reiterate that our focus is not on the specific PointPatch and DP3 architectures, which were chosen as representative examples, but on the improvement to a given method when Fourier features are applied. Unlike previous work, we demonstrate this improvement systematically across numerous tasks and architectures and show that is reliable and robust. To help make this point stronger, we will add experiments with additional point cloud architectures in variants with and without Fourier features. In the updated manuscript, we will revise Figure 5 to provide a fairer comparison between our method and the baseline and provide additional qualitative results in the form of rollout videos.
>
> Below, we provide answers to the questions raised by the reviewer:
>
> 1. Score-based diffusion, where we use the formulation called EDM presented by Karras et al.[1] is a more general framework for diffusion compared to DDPM. One of the main benefits is that score-based diffusion requires significantly fewer denoising steps than DDPM diffusion. (Compare 3D Diffuser Actor[2], which uses DDPM, against 3D FlowMatch Actor[3], an improved version using flow matching which claims a 9X speed up in inference.) Beyond this, there are a number of recent works that use EDM or score-based diffusion for imitation learning to great effect[4-5]. Since our modifications only affect the observation encoder, we expect our findings to hold for DDPM, flow matching, and other diffusion frameworks.
> 2. The paper linked by the reviewer[6] investigates training instability in consistency models, where Fourier features are used to embed the noise level. Consistency models tend to be more difficult to train than diffusion models because they are designed for single shot generation. Empirically, we have observed that training diffusion IL models with Fourier features is much more stable. Our choice of minimum wavelength was motivated by considerations of the scales of relevant objects in the scene. We will provide an ablation over number and size of wavelengths to demonstrate how the policy responds to these hyperparameters.
>
> We thank the reviewer again for their insightful questions and look forward to further discussion.
>
> [1] Karras, Tero, et al. "Elucidating the design space of diffusion-based generative models." Advances in neural information processing systems 35 (2022): 26565-26577.
>
> [2] Ke, Tsung-Wei, Nikolaos Gkanatsios, and Katerina Fragkiadaki. "3D Diffuser Actor: Policy Diffusion with 3D Scene Representations." 8th Annual Conference on Robot Learning (2024).
>
> [3] Gkanatsios, Nikolaos, et al. "3D FlowMatch Actor: Unified 3D Policy for Single-and Dual-Arm Manipulation." arXiv preprint arXiv:2508.11002 (2025).
>
> [4] Reuss, Moritz, et al. “Goal Conditioned Imitation Learning using Score-based Diffusion Policies.” Robotics: Science and Systems (2023).
>
> [5] Jia, Xiaogang, et al. “MaIL: Improving Imitation Learning with Selective State Space Models.” 8th Annual Conference on Robot Learning (2024).
>
> [6] Song, Yang, and Prafulla Dhariwal. "Improved techniques for training consistency models, 2023." *arXiv preprint arXiv:2310.14189 (2023)*.

---

> > ### Comment · Reviewer_TuVu · 2025-11-26
> >
> > I would like to thank authors for their kind responses, and some of my questions are answered. However, my major concern is still about lacking the novelty and the real-world experiment. Therefore, I would like to keep my rating.

---

> > > ### Author Response · Authors · 2025-11-28
> > >
> > > We thank the reviewer for their response and for engaging in discussion. We apologize for the delay in providing a revised manuscript, but we believe we have now been able to address many of the reviewer’s concerns.
> > >
> > > - We have added extensive real robot experiments on 3 challenging real world tasks, featuring long horizons, multiple phases, and precise manipulation. Even more than our simulation results, these real world results show that Fourier features are robust to real world noise and camera misalignment, improving the average normalized score from 5% to 35%. We direct the reviewer to chapter 5, as well as to our [project page](https://fourier-il.github.io/fourier-il/) for rollout videos.
> > > - We have added a further encoder architecture, PCM, to help back up our claim that Fourier features are a general technique that consistently improve point cloud encoders. We show that Fourier features bring PCM’s performance from 27.1% to 32.6% on the RoboCasa benchmark. While there will always be tasks where one architecture or another outperforms, we show that Fourier features provide a boost in the vast majority of cases relative to using Cartesian coordinates.
> > > - We have made the qualitative results on RoboCasa more fair by using the same scene in both cases, and have moved the figure to appendix A.3. In a coming revision, we will add further rollouts on simulation tasks to the appendix. In the meantime, we direct the reviewer to our project page for real world rollout videos, where the qualitative difference is much easier to see. Rollout videos in simulation will be added there soon.
> > >
> > > Although Fourier features are well established in other subfields of machine learning, they are yet to be widely used in 3D imitation learning. Our aim is to show their application is simple, robust to hyperparameters, and powerful. Besides strong increases in success in simulation and the real world, we also contribute extensive ablations that show that several findings from the computer vision community do not apply to imitation learning. For example, we were unable to find a benefit from learned Fourier features, and Gaussian random Fourier features significantly decreased performance. We believe these contributions are of significant value to the robotics community, but we kindly invite the reviewer to give further feedback and engage in more discussion.

---

### Official Review · Reviewer_9Bk2 · 2025-11-09

**Soundness:** 2
**Presentation:** 1
**Contribution:** 2
**Rating:** 2
**Confidence:** 4

**Summary:**

This paper investigates how Fourier features help mitigate spectral bias. The method is simple — augmenting point cloud encoders with Fourier feature embeddings to enable the network to capture high-frequency geometric cues. Experiments on several tasks from RoboCasa and ManiSkill3 demonstrate consistent improvements on both DP3 and PointPatch.

**Strengths:**

1. The idea is straightforward and reasonable at a high level. The experiments conducted in simulation convincingly demonstrate its effectiveness across two 3D baselines (PointPatch, DP3).

2. Implementation details (such as training details) are sufficient and shows good reproducibility.

**Weaknesses:**

1. Lack progressive ablation studies. The work employs VariableJitter augmentation to stabilize training with Fourier features. However, this setup lacks fair and step-by-step ablation studies. There is a critical hypothesis that VariableJitter itself may contribute to the observed improvements, or it fits Fourier features. To properly isolate the effects, the paper should include step-by-step comparisons across the following variants: baseline, baseline+aug, fourier, fourier + aug.

2. Lack direct evidence of mitigation spectral bias. The improvements should be supported by direct visualization or spectral analysis between with/wo fourier.

3. Absence of real-world experiments. Real world 3D point clouds contain more noise (e.g. point sparsity, unstable depth sensors, noise or occlusion artifacts), which could lead to unstable learning of policy or sensitivity to spurious geometry. Besides, simulation engines has coarse contact resolution compared with real-world ones. Experiments on real hardware (even under noisy or partially occluded conditions) should make the results much more convincing.

4. Limited contribution scope. The second contribution appears to be a standard validation of first contribution rather than an independent contribution.

5. Limited scope to 3D policy. The motivation regarding spectral bias should also apply to 2D inputs. The authors' justification about RGB is sensitive to viewpoint or lighting is not convincing, especially the experiments presented in the paper are not directly related to these factors. Demonstrating method effectiveness in 2D would broden the research scope.

**Questions:**

See the weakness.

---

> ### Author Response · Authors · 2025-11-20
>
> We would like to thank the reviewer for reading the manuscript and for their critical, but thoughtful review. We sincerely appreciate their effort towards improving our submission, and look forward to a productive scientific exchange in the coming weeks.
>
> The reviewer has raised the following concerns: lack of real-world experiments; lack of ablation studies, especially regarding VariableJitter; and missing spectral analysis of the learned policies to demonstrate the effect of Fourier features. In particular, the reviewer states that real world experiments are necessary to show that training with Fourier features is stable and does not result in overfitting, and we completely agree. We have initiated experiments for each of these points, and will revise our draft when the results are available.
>
> Finally, the reviewer has stated that they would like to see results on 2D inputs (e.g. RGB or pointmap modalities) with Fourier features. Preliminary experiments with Fourier features for pointmaps did not show any significant benefit so we dropped them early on. We are happy to provide experimental results on this in the appendix, as well as our hypothesis about the underlying reason. In our work, we focus on point cloud-based policies, since the advantages of point clouds over RGB for imitation learning has been demonstrated in numerous recent studies[1-4], in particular their robustness to changes in viewpoint[3] and lighting[4]. In this setting, the benefit of adding Fourier features is meaningful and robust, and therefore relevant to the broader imitation learning community. While it would be exciting to somehow extend this method to 2D inputs, we believe this is outside the scope of this work.
>
> If there are any further open questions or unresolved concerns, we encourage the reviewer to leave further comments, as we are happy to engage in further discussion.
>
> [1] Zhu, Haoyi, et al. "Point cloud matters: Rethinking the impact of different observation spaces on robot learning." *Advances in Neural Information Processing Systems* 37 (2024): 77799-77830.
>
> [2] Chisari, Eugenio, et al. "Learning Robotic Manipulation Policies from Point Clouds with Conditional Flow Matching." *Conference on Robot Learning*. PMLR, 2025.
>
> [3] Wilcox, Albert, et al. "Adapt3R: Adaptive 3D Scene Representation for Domain Transfer in Imitation Learning." *arXiv preprint arXiv:2503.04877* (2025).
>
> [4] Zhu, Yifeng, et al. "Learning Generalizable Manipulation Policies with Object-Centric 3D Representations." *Conference on Robot Learning*. PMLR, 2023.

---

> > ### Author Response · Authors · 2025-11-28
> >
> > We wish to thank the reviewer again for their constructive criticism and for their patience. We believe that the revised manuscript we just uploaded addresses many of the concerns raised in the review.
> >
> > - We have provided a detailed ablation of the effect of augmentation (random jitter or VariableJitter) in the case with and without Fourier features. The results show that the effect of VariableJitter is to reduce the variance in performance, although the choice of data augmentation does not affect the average. We direct the reviewer to chapter 6.
> > - We have added extensive real robot experiments on 3 challenging real world tasks, featuring long horizons, multiple phases, and precise manipulation. Even more than our simulation results, these real world results show that Fourier features are robust to real world noise and camera misalignment, improving the average normalized score from 5% to 35%. We direct the reviewer to chapter 5, as well as to our [project page](https://fourier-il.github.io/fourier-il/) for rollout videos.
> >
> > We are working on a visualization of the spectral responses of the learned policies, and should be able to add this to the draft within 1-2 days. We agree that this is critical for supporting our hypothesis that Fourier features mitigate the spectral bias.
> >
> > In the meantime, we welcome additional comments and criticisms from the reviewer, and look forward to a fruitful discussion.

---

### Author Response · Authors · 2025-11-20

We would like to express our heartfelt thanks to all the reviewers for taking the time to read the manuscript and formulate insightful comments. Although the job of a reviewer is often difficult and thankless, we greatly appreciate their help in improving our final manuscript.

The reviewers positively highlighted the readability, presentation[TuVu], generality[ApuY], and reproducibility[9Bk2] of the presented work, as well as the quality of the simulation experiments[9Bk2,ApuY]. The main concerns raised by the reviewers are: a lack of real-world experiments[9Bk2,TuVu,ApuY], especially to demonstrate that Fourier features work in noisy environments with imperfect camera calibration[ApuY]; a lack of ablation studies, especially on the use of jitter and VariableJitter[9Bk2] and the choice of wavelengths[ApuY]; missing spectral analysis of the learned features[9Bk2]; and more extensive simulation evaluations to demonstrate that the benefit is consistent and robust[TuVu].

In response to these concerns, we have started a number of additional experiments to shore up the manuscript:

- real-world robot experiments in noisy, multi-camera environments with and without Fourier features;
- an extensive ablation of wavelength hyperparameters and data augmentation (i.e. jitter and Variable Jitter);
- experiments with learned Fourier features and Gaussian random Fourier features;
- an investigation into the spectral response of the learned policies and how they are affected by the above ablations; and
- additional architectures and tasks to further demonstrate the robustness of our findings.

We will update the paper draft as soon as possible with these additional experimental results, after which we would graciously appreciate a further round of comments and feedback from the reviewers.

---

### Author Response · Authors · 2025-11-28
**Revised Manuscript Uploaded**

We would like to thank the reviewers once again for their valuable reviews and for their patience while waiting for the revision.

We have addressed the majority of reviewer comments and uploaded a revised version. For convenience, major changes in the text are colored orange. The most significant additions are:

- Real-world robot experiments in noisy, multi-camera environments that show the strong benefit of Fourier features in the real world [9Bk2,TuVu,ApuY]. We kindly direct the reviewers to our [anonymous project page](https://fourier-il.github.io/fourier-il) with videos of real robot rollouts with and without Fourier features.
- Extensive ablations and parameter studies of Fourier features including: the effect of random jitter or VariableJitter compared to no jitter, and how this interacts with Fourier features [9Bk2]; experiments with different numbers of wavelengths and minimum wavelength, which show that our method is remarkably robust across a range of hyperparameters [ApuY]; and comparisons against Gaussian random Fourier features and learnable Fourier features [ApuY]. These experiments validate our design choices and show the robustness of our method.
- A novel encoder architecture evaluated in simulation with and without Fourier features, which underlines the generality of our claim, namely that Fourier features are beneficial to any point cloud architecture.

Furthermore, we corrected numerous errors and clarified wording, and added the missing seeds for the Open/Close Doors tasks for the PointPatch methods. These additional seeds do not affect our findings.

Within the next 1-2 days, we will upload a further revision which will include an investigation of the spectral response of the learned encoders [9Bk2].

Before the end of the rebuttal period, we will continue to do more real world experiments with additional tasks and additional encoder architectures, to give further evidence of our method's strong performance on noisy, real world tasks. As always, we welcome further comments by the reviewers and invite another round of discussion before the end of the rebuttal period.

---

### Author Response · Authors · 2025-12-03
**Final comments and summary of discussion phase**

Dear reviewers, dear area chair,

We are deeply saddened by recent events and their repercussions for the ICLR community. For the convenience of the area chair, we have tried to summarize the rebuttal process briefly below.

With respect to the initial draft of the paper, the reviewers identified the following weaknesses, which we address as described:

- A lack of real-world experiments **[9Bk2, TuVu, ApuY]**, especially to demonstrate that Fourier features work in environments with noisy depth, point sparsity, occlusion artifacts, real-world physics, and extrinsic/intrinsic miscalibration. In response, we added real world experiments on 4 challenging, multi-phase, goal-conditioned tasks. Adding Fourier features improves our best architecture on all 4 tasks significantly and the mean normalized score from 14.8% to 40.2%. Despite the reviewers’ concerns, the policy shows no signs of overfitting to spurious geometry **[9Bk2]**. In fact, policies trained with Fourier features qualitatively have smoother movements and are more reactive to their environment, which further highlights their robustness. We encourage the reader to compare the rollout videos from each method we have uploaded to our [project page at https://fourier-il.github.io/fourier-il](https://fourier-il.github.io/fourier-il/).
- A lack of ablation studies, especially on the use of jitter and VariableJitter **[9Bk2]** and the choice of wavelengths **[ApuY]**. Reviewer **TuVu** remarked that Fourier features may be sensitive to hyperparameters. Reviewer **ApuY** asked if Gaussian random Fourier features (RFFs) or learned sinusoidal frequencies might be better than log-spaced ones. In response, we have added 3 detailed ablations: a comparison of jitter types with and without Fourier features, a comparison of different numbers and sizes of wavelengths; and a comparison with Gaussian RFFs using two different methods for learning frequencies. Our findings are: that jitter is not necessarily required to regularize Fourier features, but VariableJitter decreases the variance across seeds; Fourier features provide a remarkably stable benefit across a wide range of hyperparameters; that Gaussian RFFs hurts policy performance; and learning frequencies has no benefit and may even hurt performance. We would like to note that this runs counter to other findings, e.g. in NeRFs or image generation. Our spectral analysis provides some potential explanations for these observations (see next point).
- Reviewer **9Bk2** noted a lack of spectral analysis of the learned features, which would provide direct evidence of mitigation of spectral bias. We have analyzed the FFT spectra of the trained policies and find that adding Fourier features measurably increases the frequency response at the small wavelengths relevant for the local patch embedding. While Gaussian RFFs also achieve this, they demonstrate much lower sensitivity to large wavelengths, which is necessary for the patch position encoder to distinguish the global features of the scene. Gaussian RFFs tend to cause the patch position encoder overfit on small wavelengths. This correlates with our empirical findings on policy success.
- Reviewer **TuVu** had concerns about noisy evaluations. We added a 3rd point cloud architecture, PCM, and show that Fourier features increase its success rate from 27.1% to 32.6% on RoboCasa. With now 3 candidate architectures across 3 environment suites including the real world, we believe we have a strong claim that Fourier features are widely and generally beneficial for point cloud imitation learning.
- Reviewer **TuVu** further remarked that our qualitative comparison (originally Fig. 5) was not fair because the rollouts were on different scenes. We have moved this figure to Figure 9 in the appendix and added additional pairs of rollouts on matching scenes. For more qualitative results, we invite the reader to view our [project page at https://fourier-il.github.io/fourier-il](https://fourier-il.github.io/fourier-il/).

Lastly, we wish to emphasize that despite the fact that Fourier features are a well-known technique in other machine learning fields, we are only aware of one other imitation learning method that uses Fourier features [1], and it is not common practice to use them in supervised learning on point clouds. Since the initial draft, we have added a number of ablations and parameter studies to investigate Fourier features in the context of diffusion imitation learning and robotics. We empirically observe Fourier features to be robust to real world noise, relatively insensitive to hyperparameters, and not reliant on jitter data augmentation. Since these findings allow for a simple recipe for improving imitation learning from point clouds, we believe our findings are novel and valuable to the community.

[1] Wilcox, Albert, et al. "Adapt3R: Adaptive 3D Scene Representation for Domain Transfer in Imitation Learning." (2025)

---

### Meta-Review · Area_Chair_WnfM · 2026-01-06

**Summary:**

Most of the reviewers have concerns about novelty, contribution, real-world experiments, and noisy evaluation. These aspects are crucial for a robot learning paper. While the authors tried to add new experiments, it does not fully address the concerns.

**Reviewer Concerns:**

The real-world experiments are not systematic enough to have conclusive results. There is also no thorough ablative analysis to illustrate the scientific aspect of the method.

**Reviewer Scores:**

4 4 4 4

---

### Decision · Program_Chairs · 2026-01-26

Reject